# Surface Modification of Carbon Fibers by Grafting PEEK-NH2 for Improving Interfacial Adhesion with Polyetheretherketone

**DOI:** 10.3390/ma12050778

**Published:** 2019-03-07

**Authors:** Elwathig. A. M. Hassan, Tienah. H. H. Elagib, Hafeezullah Memon, Muhuo Yu, Shu Zhu

**Affiliations:** 1Key Laboratory of High-Performance Fibers & Products, Ministry of Education, College of Materials Science and Engineering, Donghua University, Shanghai 200051, China; elwathig2011@uofg.edu.sd (E.A.M.H.); solarahussain@yahoo.com (T.H.H.E.); yumuhuo@dhu.edu.cn (M.Y.); 2Key Laboratory of Shanghai City for lightweight composites, Donghua University Center for Civil Aviation Composites, Donghua University, Shanghai 200051, China; hafeezullah_m@yahoo.com; 3State Key Laboratory for Modification of Chemical Fibers and Polymer Materials, Donghua University, Shanghai 200051, China; 4Industries Engineering and Technology, University of Gezira, Wad Madani 21111, Sudan; 5Key Laboratory of Textile Science & Technology, Ministry of Education, College of Textiles, Donghua University, 2999 North Renmin Road, Shanghai 201620, China

**Keywords:** carbon fiber (CF), polyetheretherketone (PEEK), aminated polyetheretherketone (PEEK-NH_2_), interfacial adhesion

## Abstract

Due to the non-polar nature and low wettability of carbon fibers (CFs), the interfacial adhesion between CFs and the polyetheretherketone (PEEK) matrix is poor, and this has negative effects on the mechanical properties of CF/PEEK composites. In this work, we established a modification method to improve the interface between CFs and PEEK based chemical grafting of aminated polyetheretherketone (PEEK-NH_2_) on CFs to create an interfacial layer which has competency with the PEEK matrix. The changed chemical composition, surface morphology, surface energy, and interlaminar shear strength were investigated. After grafting, the interlaminar shear strength (ILSS) was improved by 33.4% due to the covalent bonds in the interface region, as well as having good compatibility between the interface modifier and PEEK. Finally, Dynamic Mechanical Analysis (DMA) and Scanning Electron Microscopy (SEM) observation also confirmed that the properties of the modified CF/PEEK composites interface were enhanced. This work is, therefore, a beneficial approach towards enhancing the mechanical properties of thermoplastic composites by controlling the interface between CFs and the PEEK matrix.

## 1. Introduction

As lightweight materials, carbon fiber/polyetheretherketone (CF/PEEK) composites have been used in a broad range of applications with a promising future in automobiles, aerospace, military defense, biomedicine, and the electronic industry, owing to their low density, high specific strength, and high thermal stability [1,2,3,4]. Previous studies have proved that the effective mechanical properties of the composites depend on the properties of each of the components as well as on the fiber/matrix interface quality [5]. However, PEEK exhibits some drawbacks such as poor adhesion of hydrophilic reactive groups in PEEK chains [6,7]. On the other hand, due to its non-polar surface and compound of highly crystallized graphitic basal planes with inert structures, the pristine CFs have poor interaction with most of the polymers [8]. For these reasons, their applications have been limited because of weak fiber-matrix adhesion, which causes premature failure of the composites [9], thus, encouraging improvements in their performance for more demanding technological applications. Therefore, varieties of methods regarding surface modifications of CFs were established to improve the interfacial adhesion of composites, such as oxidation treatment, plasma, and ozone treatment [10,11,12,13,14,15]. However, most of them were established for improving interfacial adhesion between CFs and thermosetting polymers because of strong chemical interactions between modified CFs and thermosetting polymers. In the case of thermoplastic polymers, it is difficult for the functional groups provided by thermoplastics to react chemically with the surface functional groups of CFs. Therefore, only a little improvement can be achieved by increasing surface roughness or interacting with oxygen-containing functional groups. A polymer as compatibilizer that has a similar, or the same main component as the matrix, is desired to promote interfacial adhesion of the composites [16]. Grafting of the compatibilizer of the modified PEEK creates a strong interface layer on CFs, which improves interfacial adhesion between fiber and the matrix by way of a “bridging” effect. The compatibilizer either interacts physically or chemically with the fiber surface or has an excellent compatibility with the polymer matrix. Yuan et al. [17] applied polyethersulfone (PES) emulsion sizing to the CF surface, and the interfacial strength of CF/PES composites was successfully increased by 26%. Liu et al. [18] improved interfacial adhesion of CF/poly(phthalazinoneether ketone) (PPEK) by coating it in a PPEK film. Based on the discussions above, it can be maintained that grafting matrix-compatible resin on the CF surface may produce a synergistic effect on improving the interfacial properties of composites.

In this study, low contents of PEEK with plentiful amine groups on its molecular chains were chosen as the modifiers and were grafted onto CFs by covalent linkage. It is expected that those physical and chemical bonds, as well as molecular chain entanglement between modified PEEK and the PEEK matrix, are beneficial to the improvement of interfacial interactions.

## 2. Materials and Methods

### 2.1. Materials

CF fabrics (3K-T300-plain) were provided by Toray Industries, Inc. (Tokyo, Japan). PEEK (1000-300G) was supplied in film form by Victrex (Lancashire, UK). Acetone (≥99.5%) was received from Shanghai Yunli Economic and Trading Co., Ltd. (Shanghai, China). Dimethylacetamide (DMA) was provided from Shanghai Ling Feng Chemical Reagent Co., Ltd. (Shanghai, China).

### 2.2. Amination of PEEK

A stirred suspension of PEEK in HNO_3_/H_2_SO_4_ was heated under the conditions specified in Table 1. The product (PEEK-NO_2_) was filtered and washed with water until pH 7 was obtained, then it was washed with ethanol, acetone and dried in an oven at 80 °C. SnCl_2_ was dissolved in HCl solution (37%) and then ethanol (50 mL) was added into the mixture. After stirring at 65 °C for 15 min, the prepared PEEK-NO_2_ powder was slowly poured in and the reaction was kept for 4 h at 65 °C. Then the yellowish precipitate was filtered off and washed with distilled water. Followed by drying in an oven for 24 h, yellowish PEEK-NH_2_ was prepared (Figure 1).

### 2.3. Grafting PEEK-NH2 onto CFs

The CFs were refluxed in acetone for 48 h at 70 °C to remove sizing agents, then they were washed repeatedly in deionized water and dried at 80 °C to obtain the desized CF. Subsequently, the CFs were treated with Meldrum acid solution at 30 °C for 3 h followed by washing with acetone and drying. The obtained CFs were denoted as activated-CFs (ACFs). The activated CFs were tied to a glass frame and placed into PEEK-NH_2_/ dimethylformamide (DMF) solution with a different degree of amination, then the reaction was performed under nitrogen atmosphere at 50 °C for 48 h to induce plenty of PEEK-NH_2_ onto the CF surface. The samples were denoted as PEEK-NH_2_-1@CF, PEEK-NH_2_-2@CF, PEEK-NH_2_-3@CF, and PEEK-NH_2_-4@CF. The overall reaction is shown in Figure 2.

### 2.4. Preparation of Modified CF/PEEK Resin Composites

The desized, modified CF and PEEK films were dried in a vacuum oven at 100 °C for 24 h before the manufacturing procedures of composites took place. The laminates were manufactured by alternatively placing 8 plies of CF fabrics and 9 plies of PEEK films with the transverse dimensions of 200 mm × 150 mm. The compression molding method was used to prepare CF/PEEK composite panels at 390 °C under the pressure of 2.5 MPa for 25 min. Finally, the laminates were cooled to 100 °C with a cooling rate of 5 °C/min by keeping a constant pressure (2.5 MPa). The laminates had a matrix volume fraction of 60%, a fiber volume fraction of 40%, and an average void content <2.5%.

## 3. Characterizations

Morphologies of modified CF and fractural morphologies of CF/PEEK composites were observed using a scanning electron microscope (SEM) (HITACHI S-300N, Tokyo, Japan) with an acceleration voltage of 15–18 kV. Solid-state NMR experiments were performed on a Bruker DMX-400 spectrometer operating at a ^l3^C frequency of 100.62 MHz. The functional groups of modified PEEK and PEEK grafted CFs were identified by a Fourier transform infrared (FT-IR) spectrometer (Nicolet 8700, USA) using powder-pressed KBr disks in wave numbers ranging from 400 cm^−1^ to 4000 cm^−1^. Thermogravimetric analysis (TGA) was performed under air atmosphere using a TGA Q5000 IR (TA Instruments-Waters LLC, New Castle, DE, USA) with a heating rate of 10 °C/min from room temperature to 800 °C. The contact angle analysis test (OCA40Micro, Germany) was used to determine the surface energies of modified CF and the polar components of surface energies were calculated according to the Wilhelmy method [19]. Five measurements were performed at different locations for each kind of CF.

Dynamic mechanical analysis (DMA) was carried out using a dynamic mechanical thermal analyzer (TA Q800, New Castle, DE, USA) under a three-point bending mode. The samples were tested at a frequency of 1.0 Hz from 30 °C to 290 °C, at a heating rate of 10 °C/min. Interlaminar shear strength (ILSS) of composites were carried out on a universal testing machine (LABSANS LD26.105, China) according to ASTM D7264. Five parallel measurements were conducted and averaged for each final result. The standard deviation was indicated by error bars.

## 4. Results and Discussion

### 4.1. Surface Morphologies of Modified CFs

As shown in Figure 3, PEEK-NH_2_ were distributed on the CF surface, and the amount of amine-PEEK depended on the degree of amination, confirming that the grafting of PEEK-NH_2_ increases with the increasing degree of amination. For PEEK-NH_2_-4@CF, the fiber uniformly wrapped by particles of PEEK (illustrated by arrows) indicated that PEEK-NH_2_ was chemically grafted onto CFs. It can be seen in Figure 3D, that a layer of uniform PEEK particles on the fiber surface was generated, which served as a bridge to connect CF and the PEEK matrix at the composite interface region, and consequently enhanced the interfacial adhesion between CFs and the matrix.

### 4.2. Surface Energies of Modified CFs

Polar liquid (water) and non-polar liquid (glycerol) was used to determine the dispersive (γ^d^) and polar (γ^p^) components of surface energy according to the Wilhelmy method [19]. Surface polar functional groups of CFs enhanced the polar component, while the topography of fiber dominated the dispersive component. Therefore, the enhanced polar component of surface energy can be interpreted as many polar amino groups induced onto the surface of grafted CFs. It was found that an increase in total surface energy and its polar component appears to be due to the increase in the percentage of surface polar functional groups (NH_2_). In fact, an approximate linear relationship can be obtained between the polar component of the surface energy, and polar functional groups on the surface of CFs (see Table 2). The improved dispersive component of different CF surfaces can be ascribed to increased fiber surface roughness caused by the polymer sizing agent containing PEEK-NH_2_ which was grafted on the fiber surface. The improvement of carbon fiber surface energy effectively improved the interfacial bonding between fiber and polymer matrices.

### 4.3. Thermal Stability of Modified CFs

As shown in Figure 4, the initial degradation temperature of desized@CF was about 640 °C [20]. Meanwhile, there was a significant weight loss for CF grafted with PEEK-NH_2_ which showed two distinct weight losses at 480 °C and 640 °C. It can be seen that with increasing amination degree the thermal degradation continuously increased which was attributed to reactions with the main chain and derivatives of the amino group. This could have affected the level of the improvement, especially at high processing temperature. The results of the thermal analysis for all investigated samples are summarized in Table 3. It is well-known that the amino group of PEEK can form interaction bonds with basic groups in PEEK, thus, leading to the stabilization of the aromatic of the amino group to some extent.

### 4.4. The Surface Chemical Elemental Composition of CFs

#### 4.4.1. NMR of Modified PEEK

The reaction of the functional polymer with -NO_2_ was investigated with NMR. The ^13^C NMR spectrum of PEEK and PEEK-NO_2_ are given in Figure 5. The polymer backbone of PEEK peaks resonated at 120.2, 132.5, 150.8, 160, and 193 ppm. These peaks shifted to lower chemical shifts in the functional polymer (PEEK-NO_2_) compared with that of PEEK, and a new peak at 142.4 ppm was observed, which was attributed to the –NO_2_ grafted onto the PEEK main chain [21].

#### 4.4.2. FTIR Analysis of Modified PEEK

Figure 6A shows the infrared spectra for PEEK and nitrated PEEK with different nitration degrees, and the absorption bands observed at 1543 cm^−1^, and 1346 cm^−1^ relate to the unsymmetrical and symmetrical stretching of the nitro group, respectively [21]. The appearance of a band at 3434 cm^−1^ related to N-H stretching, which increased with the increasing degree of nitration, and the peaks at 928 cm^−1^ were assigned to C-N stretching. For PEEK-NH_2_, as shown in Figure 6B, the peaks at 3474 cm^−1^, 3374 cm^−1^, and 1651 cm^−1^ were attributed to N-H stretching, the peak at 1306.3 cm^−1^ was attributed to C-N stretching, and the peak at 1534 cm^−1^ was attributed to N=N stretching and the intensity of this group increased with amination degree increases [22]. All these changes indicated that the nitro groups were reduced to amino groups. For activated CFs (Figure 6C), peaks observed at 3439 cm^−1^, 1753 cm^−1^, and 1118 cm^−1^ were associated with stretching vibrations of -OH groups, C=O groups, and -COOH groups, respectively. For the PEEK-NH_2_ grafted CFs, the disappearance of the peak located at 1710 cm^−1^ verified the alteration of the molecular structure of the carboxylic acid group [23]. The peak at 1531 cm^−1^ corresponded to the bending vibration of the N–H of amide, indicating that PEEK-NH_2_ was grafted onto the fiber surface through covalent bonds [24].

### 4.5. Mechanical Properties of Modified CFs

The typical stress-strain curves obtained from the ILSS tests of modified CF/PEEK composites are presented in Figure 7A. There was a slight increase in the maximum load of the ILSS test with the increasing amination degree of the modifier, and then it decreased when the amination degree was further increased. The inset of Figure 7A shows the morphologies of failure modes observed in the CF/PEEK laminates after ILSS tests were carried out. The fracture surfaces of specimen after ILSS tests were delamination cracked, and located at the critical ply interface. ILSS of CF/PEEK composites modified by PEEK-NH_2_ increased compared with that of the desized@CF composite as shown in Figure 7B. The results suggest that the compatibilizer PEEK-NH_2_ influenced the mechanical properties of CF/PEEK. For PEEK-NH_2_-1@CF and PEEK-NH_2_-2@CF, the amino groups were not enough to be attached uniformly onto the CF surface, thus, the fiber surface was not totally covered by the grafting material. Hence, ILSS showed only a slight improvement. For PEEK-NH_2_-3@CF, there were enough active groups (e.g., NH_2_) in the PEEK. Therefore, the amino of PEEK-NH_2_ also had the chemical reaction and intermolecular force of COOH groups of activated CFs, and thus, gave uniform polymer grafting. Hence, improvement in ILSS for PEEK-NH_2_-3@CF reached a maximum, i.e., 33.4%. In addition, the molecular backbone of PEEK-NH_2_ was similar to that of PEEK. Thus, PEEK and PEEK-NH_2_ could have good physical compatibility. These results demonstrated a significantly higher level of improvements than were reported in previous studies [15,25,26]. But compared with our previous study [2], there was a limited level of improvement, which could be attributed to the sensitivity of the amino group to high temperatures. On the other hand, when increasing the amination degree, as in PEEK-NH_2_-4@CF, the ILSS tended to decrease due to lower thermal stability. Because of the lower degradation temperature (Figure 4), the interfacial layer of this sample might have quickly degraded and formed a weak interfacial layer during the processing of CF/PEEK composites at high temperatures.

### 4.6. Dynamic Mechanical Analysis of Modified CFs

The variation in the storage modulus (E’) as a function of temperature for the PEEK-NH_2_ modified composites is given in Figure 8. There was a clear increase in E’ after modification of CFs with PEEK-NH_2_, and the maximum E’ values were found for the PEEK-NH_2_-3@CF composite. This indicates that the CF/PEEK interaction adhesion improved due to the reaction between the amino groups of PEEK, and COOH groups of CF. That is to say, the compatibilizer PEEK-NH_2_ was likely to have a good effect on the interfacial adhesion between the PEEK matrix and CFs. However, due to the high processing temperature of CF/PEEK composites, PEEK-NH_2_ led to more degradation with an increased degree of amination, which was reflected in decreased storage modulus probably due to damage of the interfacial layer in CF/PEEK composites.

The tanδ of the composites mainly depended on the interface between fiber and matrix. A good interface bore greater stress and less energy dissipation. However, composite materials with poor interfacial bonding tended to dissipate more energy showing a high magnitude of damping peak in comparison to a material with a strongly bonded interface. This shows that the peak value of tanδ in the modified composites with PEEK-NH_2_ was lower than that of CF/PEEK without PEEK-NH_2_. It was suggested that the interfacial compatibility of CF/PEEK composite with PEEK-NH_2_ was improved. However, the value of tanδ was affected by the degree of amination due to the high sensitivity of NH_2_ due to high temperature.

### 4.7. The Fractured Surface of the Modified Composite

The pulled out CF and the big crack in the interface of the desized CF/PEEK composite were observed (see arrows in Figure 9Ai,ii). As shown in Figure 9Aiii, the debonding failure surfaces presented a little matrix attached to the CF surface. These results indicated a poor interface between CF and PEEK. The surface morphologies of PEEK-NH_2_-1@CF are shown in Figure 9Bi,ii. The pulled out CFs in the modified CF/PEEK composite were not observed but a small crack was deflected. As shown in Figure 9Biii, some PEEK resin detached from the CFs surface owing to the relatively weak adhesion between CFs and PEEK resin. For PEEK-NH_2_-2@CF, the crack was reduced and there was no gap between CF and the matrix (Figure 9Ci,ii). Furthermore, as shown in Figure 9Ciii, more PEEK resin was attached in CFs compared with PEEK-NH_2_-1@CF, which indicated that the interface improved to some extent. For PEEK-NH_2_-3@CF, it suggested that the fiber was broken together with the matrix during the fracture, which further substantiates adhesion between PEEK and CF (Figure 9Di,ii). Besides; it is found that more resins completely adhered on the surface of PEEK-NH_2_-3@CF (Figure 9Diii). It can be inferred that the interfacial properties of PEEK-NH_2_-3@CF composites were improved with the increased degree of amination, which is in good agreement with previous experimental results of ILSS. The improvement can be attributed to the chemical bond between PEEK-NH_2_ and COOH in CF, thus, building an interfacial layer on the surface of CF while improving wettability with the PEEK matrix. Meanwhile, the compatibilizer also had good compatibility with the PEEK matrix.

## 5. Conclusions

In this work, a modified method based on the PEEK-NH_2_ matrix was investigated. The PEEK-NH_2_ was grafted onto CFs via chemical bonding. Effective improvement in the interfacial adhesion could be achieved by forming a covalent bond between the amino group of PEEK and COOH of the activated CF. Although, PEEK-NH_2_ has good compatibility with PEEK, the study showed a limited influence on improving interfacial adhesion, i.e., 33.4% for ILSS, probably due to the high sensitivity of the amine group to high temperatures. Meanwhile, the compatibility effect of PEEK-NH_2_ in the PEEK/CF composite was proved not only by the mechanical properties, but also by the storage modulus and SEM observation of modified composites, since the properties of the CF/PEEK interface were significantly enhanced. This approach may be applied to other thermoplastics that have lower processing temperature to enhance their performance.

## Figures and Tables

**Figure 1 materials-12-00778-f001:**
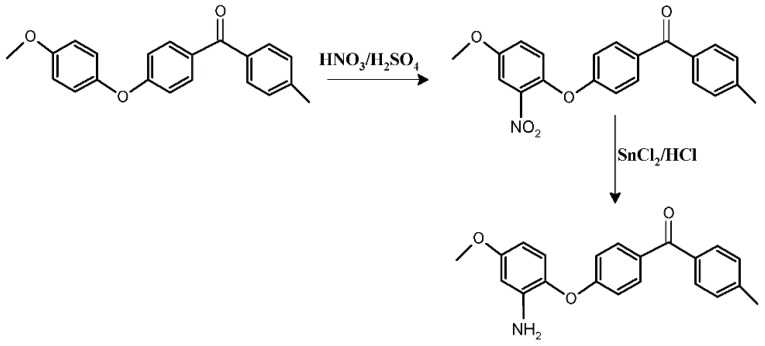
Amination of polyetheretherketone (PEEK).

**Figure 2 materials-12-00778-f002:**
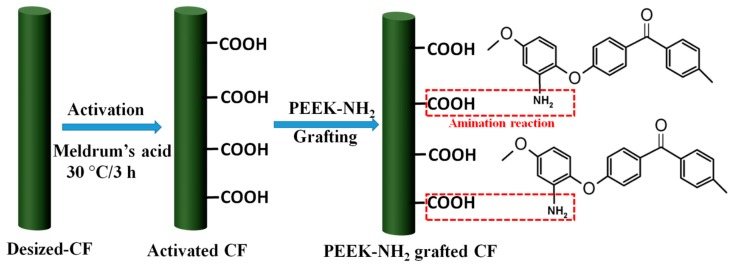
Complete representation of the grafting procedures.

**Figure 3 materials-12-00778-f003:**
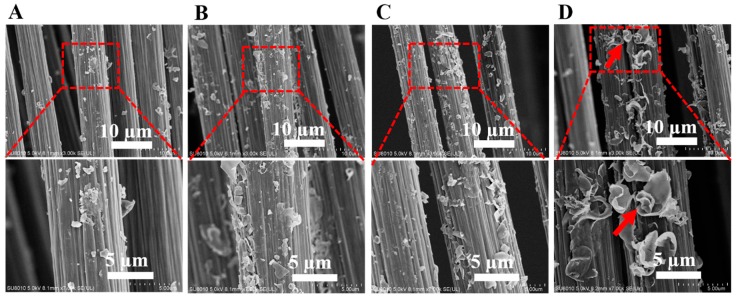
SEM images of (**A**) PEEK-NH_2_-1@CF, (**B**) PEEK-NH_2_-2@CF, (**C**) PEEK-NH_2_-3@CF, and (**D**) PEEK-NH_2_-4@CF.

**Figure 4 materials-12-00778-f004:**
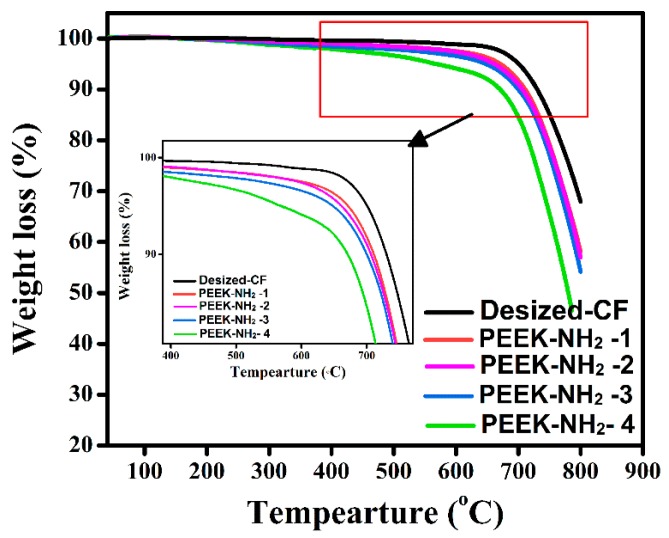
Thermogravimetric analysis (TGA) of different degrees of amination of PEEK.

**Figure 5 materials-12-00778-f005:**
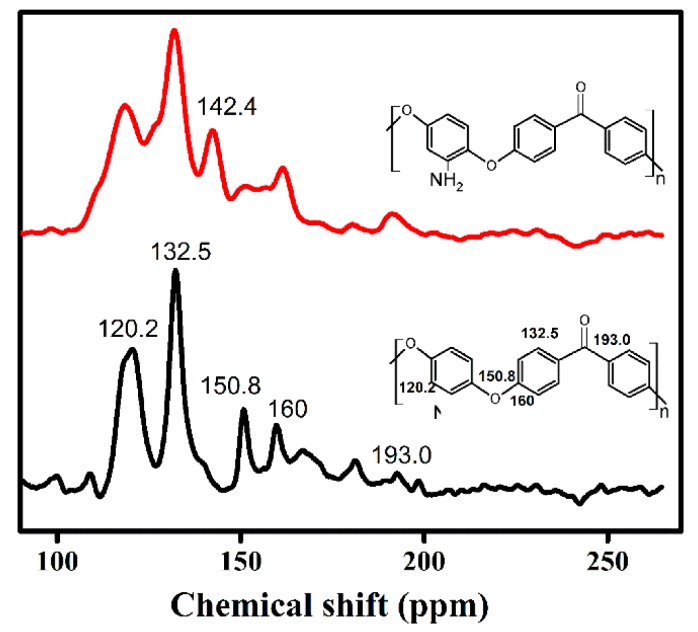
^13^C NMR of PEEK and PEEK-NO_2_.

**Figure 6 materials-12-00778-f006:**
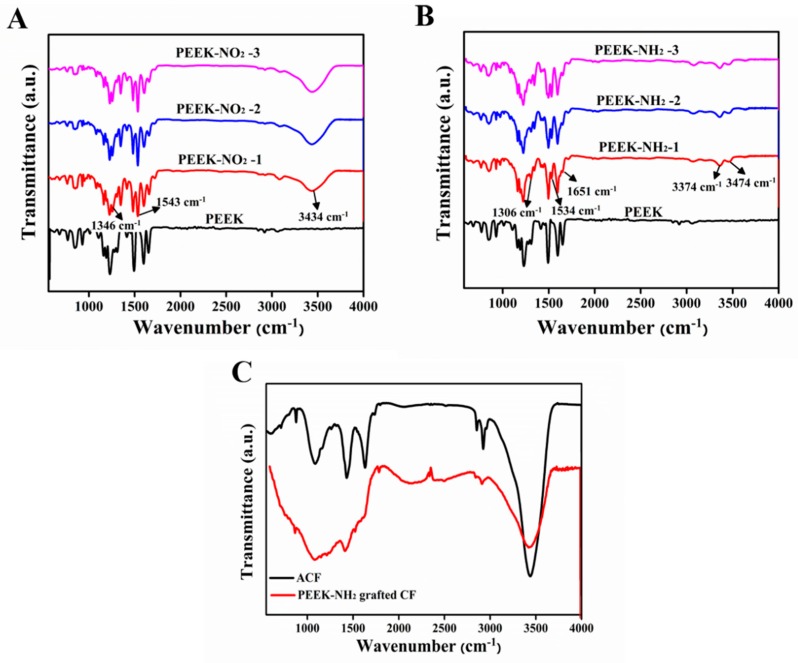
FTIR spectra (**A**) nitrided PEEK, (**B**) aminated PEEK, and (**C**) activated-CF (ACF) and PEEK-NH_2_ grafted CF.

**Figure 7 materials-12-00778-f007:**
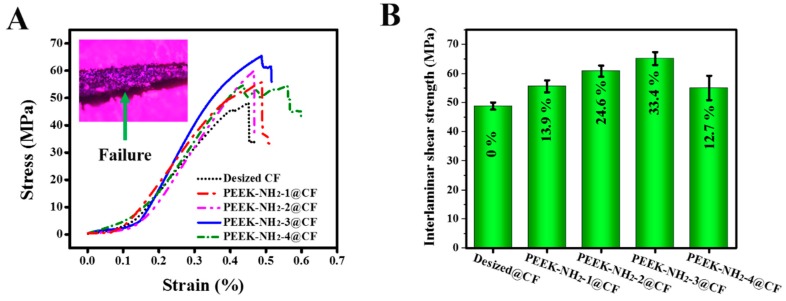
(**A**) Stress–strain curves in interlaminar shear strength (ILSS) tests, (**B**) ILSS at different degrees of amination of PEEK.

**Figure 8 materials-12-00778-f008:**
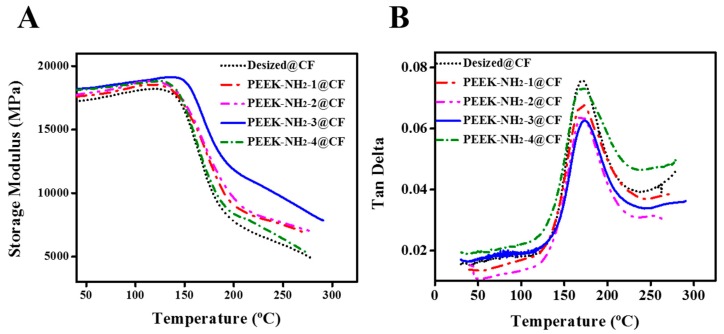
(**A**) Storage modulus and (**B**) tan δ of PEEK-NH_2_ modified CF/PEEK composites.

**Figure 9 materials-12-00778-f009:**
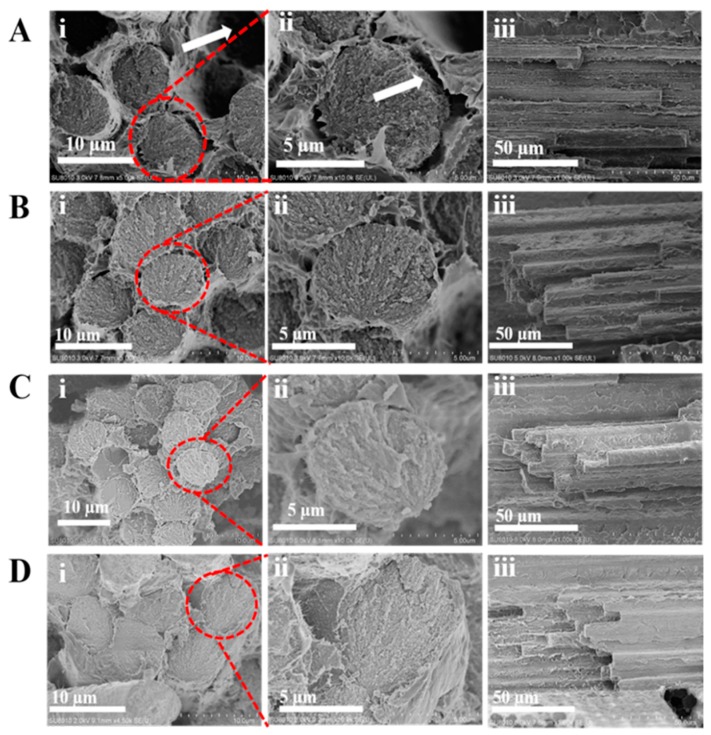
SEM images of fracture surface micrographs of CF-PEEK composites containing CF with different modifications: (**A**) Desized@CF, (**B**) PEEK-NH_2_-1@CF, (**C**) PEEK-NH_2_-2@CF, and (**D**) PEEK-NH_2_-3@CF.

**Table 1 materials-12-00778-t001:** Conditions of the nitration reaction of the products.

Batch	T/(°C)	Reaction Time/(min)
PEEK-NH_2_-1	50	30
PEEK-NH_2_-2	50	60
PEEK-NH_2_-3	50	90
PEEK-NH_2_-4	50	120

**Table 2 materials-12-00778-t002:** Surface energies of modified carbon fibers (CFs).

Samples	Contact Angle (°)		γ^d^ (mJm^−2^)	γ^p^ (mJm^−2^)	γ (mJm^−2^)
	Water	Glycerol			
Desized-CF	79.9	92.4	1.03	44.64	45.67
PEEK-NH_2_-1@CF	77	81	1.50	64.92	66.42
PEEK-NH_2_-2@CF	74	78.2	1.62	70.48	72.10
PEEK-NH_2_-3@CF	71	75.2	1.92	77.52	79.44
PEEK-NH_2_-4@CF	68	73	1.67	79.71	81.38

**Table 3 materials-12-00778-t003:** Thermal properties of modified CFs grafted with PEEK-NH_2._

Sample	T_5_ (°C)	Char Yield (%)
**Desized-CF**	700	68
**PEEK-NH_2_-1@CF**	674	59
**PEEK-NH_2_-2@CF**	670	57
**PEEK-NH_2_-3@CF**	647	54
**PEEK-NH_2_-4@CF**	568	46

T_5_: Temperature at 5% weight loss in air. Char yield (%): Residual weight at ~800 °C in air.

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
