# Peer review of "Surface Modification of Carbon Fibers by Grafting PEEK-NH2 for Improving Interfacial Adhesion with Polyetheretherketone"

_materials, 2019, doi:10.3390/ma12050778_

Round 1
Reviewer 1 Report
The paper was quite well writen, the furface of CFs was modificated to aimprove the interfical adhesion with PEEK. Before acceptance, several points should be considered:
1.Line 112, the language should be checked.
2.LIne 114, 'ILSS' should be firstly defined.
3.LIne 146, there was a mistake in spelling.
4.LIne 178, 'ACF' should be firstly defined.
5. Line 186, the language should be checked.
6. LIne 198, there should be a 'a' before 'limit level'
7.Figure 7 and 8, the color for describing PEEK-3 and PEEK-4 should be easily distingguished. At this moment, the both colors were too similar.
8.Figure 9, there is intestingly no figures for PEEK-NH2-4@CF. Please explain.
9. From line 258 to line 268, please carefully check the content about Author contribution and Funding.
Author Response
Response to Reviewer 1 comments
We sincerely thank the reviewer for constructive criticisms and valuable comments, which were of great help in revising the manuscript.
Point 1.Line 112, the language should be checked.
Response 1: Grammar error in the sentence has been corrected (Dynamic mechanical analysis (DMA) was carried out using dynamic mechanical thermal analyzer (TA Q800, USA) under three-point bending mode).
Point 2.Line 114, 'ILSS' should be firstly defined.
Response 2: We apologize for this error, ILSS firstly defined in abstract section.
Point 3.Line 146, there was a mistake in spelling.
Response 3: We are sorry for our carelessness, we have corrected the spelling mistake.
Point 4.Line 178, 'ACF' should be firstly defined.
Response 4: As recommended by the reviewer, ACF firstly defined in grafting PEEK-NH2 onto CFs section
Point 5. Line 186, the language should be checked.
Response 5: Grammar errors have been carefully corrected (The fracture surfaces of the specimen after ILSS tests were delamination crack which located at the critical ply interface).
Point 6. Line 198, there should be a 'a' before 'limit level'
Response 6: We apologize for this mistake, the article has been added accordingly.
Point 7.Figure 7 and 8, the color for describing PEEK-3 and PEEK-4 should be easily distinguished. At this moment, both colors were too similar.
Response 7: We agree with the reviewer that the color for describing samples should be easily distinguished to add relevance to the readers. We have replaced Figure 7 and 8 with new figures include different line style.
Point 8.Figure 9, there is intestingly no figures for PEEK-NH2-4@CF. Please explain.
Response 8: The authors are thankful for reviewers for highlighting the need for SEM image for PEEK-NH2-4@CF, we completely agree with your idea. However, considering the negative trend in the ILSS (shown in Figure 7. B), and dynamic mechanical analysis (Figure 8), the authors believe not to present SEM image into the final manuscript.
Point 9. From line 258 to line 268, please carefully check the content about Author contribution and Funding.
Response 9: These parts also have been carefully checked and presented.

Reviewer 2 Report
The author did appreciable work, and I have some point for the author consideration
1- In characterization part page 3, can you mention TGA acronym, TGA parameters like heating rate, used gas, etc. please?
2- In page 5 line 145 the sentence “weight loss stages at 380 ºC and 640 ºC”. However, am not seeing any weight loss at 380 C.
3- The same TGA, what the char yield for each condition?
4- It mentioned that “initial degradation temperature of desized@CF was about 640 ºC. In my opinion, this degradation temperature is too high. If you think is not please include the citation for people with the same observation.
5- In page 6, section 4.4.2, Chemical group analysis, I prefer to replace it with FTIR, since the previous section is NMR . I think NMR is Chemical group analysis also, so I prefer, either combine all chemical group analysis in one tittle or make it many sections like:NMR,FTIR, etc.
6- In page 7 in mechanical properties is not mentioned how many samples done for each parameter.
7- In page 7, line 199-200 ‘the 199 ILSS tend to be decreased due to lower thermal stability’. is the test done with different temperatures? Explain please
8- What is y-axis for figure 8-B?
Author Response
Response to Reviewer 2 comments
Thank you very much for your minute observation and valuable comments. We strongly believe that the comments and suggestions could increase the scientific value of the revised manuscript by many folds.
Point 1. In characterization part page 3, can you mention TGA acronym, TGA parameters like heating rate, used gas, etc. please?
Response 1: Thermogravimetric analysis (TGA) was performed under air atmosphere using a TGA Q5000 IR (TA Instruments- waters LLC) with a heating rate of 10 °C/min from room temperature to 800 °C.
Point 2. In page 5 line 145 the sentence “weight loss stages at 380 ºC and 640 ºC”. However, am not seeing any weight loss at 380 ºC.
Response 2: We rechecked the thermal properties result and just found a simple mistake in weight loss stages for Figure 4, which has now been rectified.
Point 3. The same TGA, what the char yield for each condition?
Response 3: We thank the reviewer for bringing up this point. We have added the char yield for each sample in section 4.3 (Table 3). For your convenience, we have also listed them below.
Table 3. Thermal properties of modified CFs grafted with PEEK-NH2
Sample | T5 (ºC) | Char yield (%) |
Desized-CF | 700 | 68 |
PEEK-NH2-1@CF | 674 | 59 |
PEEK-NH2-2@CF | 670 | 57 |
PEEK-NH2-3@CF | 647 | 54 |
PEEK-NH2-4@CF | 568 | 46 |
T5: the temperature at 5% weight loss in air and Char yield (%): residual weight at ~800 C in air
Point 4. It mentioned that “initial degradation temperature of desized@CF was about 640 ºC. In my opinion, this degradation temperature is too high. If you think is not please include the citation for people with the same observation.
Response 4: We are grateful for the suggestion. The reference below have been cited into the manuscript which showed that initial degradation temperature of desized@CF was ~ 640 ºC
Cuiqin, F.; Jinxian, W.; Julin, W.; Tao, Z., Modification of carbon fiber surfaces via grafting with Meldrum's acid. Appl. Surf. Sci. 2015, 356, 9-17.
Point 5. In page 6, section 4.4.2, Chemical group analysis, I prefer to replace it with FTIR, since the previous section is NMR. I think NMR is Chemical group analysis also, so I prefer, either combine all chemical group analysis in one title or make it many sections like NMR, FTIR, etc.
Response 5: Thank you so much for your suggestion. Yes, we have changed the section 4.4.2 into FTIR analysis of modified PEEK for better understanding.
Point 6. In page 7 in mechanical properties is not mentioned how many samples done for each parameter.
Response 6: Per your comment, we added two sentences at the end of section 3 (see below).
Five parallel measurements were conducted and averaged for each final result. The standard deviation was indicated by error bars.
Point 7. In page 7, line 199-200 ‘the 199 ILSS tend to be decreased due to lower thermal stability’. Is the test done with different temperatures? Explain, please.
Response 7: Thanks for the reviewer's perfect question. This test performed at room temperatures. In our opinion, during the processing of CF/PEEK composites at high temperatures, because of lower degradation temperature (Figure 4), the interfacial layer of this sample might quickly degrade and form a weak interfacial layer which could affect in inter-laminar shear strength.
Point 8. What is y-axis for figure 8-B?
Response 8: We are sorry for our carelessness. We have replaced it with a new figure that can show tan delta as y-axis in Figure 8-B.

Reviewer 3 Report
The above mentioned article is well written and provide scientific information to readers pertaining to surface modifications for carbons for improving adhesion between CFs and PEEK matrix. The research study presents good findings and has developed a mechanism to improve the surface morphologies of the CFs.
I would like to presents the following suggestions to the authors to improve the publication.
The introduction section only talks about few research studies that are previously accomplished in this field. It would be better for the readers if authors could put few more references to the previous research studies.
In figure 3 D, Please mark or indicate the uniform PEEK particles layer on the fiber surface with arrow.
Please re-write ..the Sentence on line 144-145 "Meanwhile, the weight loss of the PEEK-NH2 grafted CF significantly decreases and exhibits two 145 distinct weight loss stages at 380 ºC and 640 ºC."
Check the spelling of word "continuo" on line 146.
The results presented in figure 7 A &B are averaged results of all the tested samples or just a typical response of one specimen. Please make suitable correction in text to add this information.
Author Response
Response to Reviewer 3 comments
We would like to thank the reviewer for a careful and thorough reading of this manuscript and for the thoughtful comments and constructive suggestions, which help to improve the quality of this manuscript.
Point 1. The introduction section only talks about few research studies that are previously accomplished in this field. It would be better for the readers if authors could put few more references to the previous research studies.
Response 1: We very much appreciate the reviewer’s suggestion and we have updated literature review with the addition of more previous studies related to this field.
1. Luo, H.; Xiong, G.; Yang, Z.; Raman, S. R.; Li, Q.; Ma, C.; Li, D.; Wang, Z.; Wan, Y., Preparation of three-dimensional braided carbon fiber-reinforced PEEK composites for potential load-bearing bone fixations. Part I. Mechanical properties and cytocompatibility. J.Mech.Behav. Biomed. Mater. 2014, 29, 103-113.
2. Takei, H.; Salvia, M.; Vautrin, A.; Tonegawa, A.; Nishi, Y., Effects of Electron Beam Irradiation on Elasticity of CFRTP (CF/PEEK). Mater. trans. 2011, 52, (4), 734-739.
3. Li, J., Interfacial studies on the ozone and air-oxidation-modified carbon fiber reinforced PEEK composites. Surf. Interface Anal. 2009, 41, (4), 310-315.
4. Jang, J.; Kim, H., Improvement of carbon fiber/PEEK hybrid fabric composites using plasma treatment. Polym. Compos. 1997, 18, (1), 125-132.
5. Ashrafi, B.; Díez-Pascual, A. M.; Johnson, L.; Genest, M.; Hind, S.; Martinez-Rubi, Y.; González-Domínguez, J. M.; Martínez, M. T.; Simard, B.; Gómez-Fatou, M. A., Processing and properties of PEEK/glass fiber laminates: Effect of addition of single-walled carbon nanotubes. Composites Part A. 2012, 43, (8), 1267-1279.
6. Pan, L.; Yapici, U., A comparative study on mechanical properties of carbon fiber/PEEK composites. Adv. Compos. Mater. 2016, 25, (4), 359-374.
Point 2. In figure 3 D, Please mark or indicate the uniform PEEK particles layer on the fiber surface with arrow.
Response 2: We have followed the reviewer’s suggestion; the uniform PEEK particles layer have marked with arrows.
Point 3. Please re-write the sentence on line 144-145 "Meanwhile, the weight loss of the PEEK-NH2 grafted CF significantly decreases and exhibits two distinct weight loss stages at 380 ºC and 640 ºC."
Response 3: According to the reviewer’s comment, we have revised the sentence as “Meanwhile, there was a significant weight loss for CF grafted with PEEK-NH2 which showed two distinct weight losses at 480 ºC and 640 ºC.”
Point 4. Check the spelling of word "continuo" on line 146.
Response 4: We apologize for this mistake, we have corrected the spelling mistake.
Point 5. The results presented in figure 7 A &B are averaged results of all the tested samples or just a typical response of one specimen. Please make suitable correction in text to add this information.
Response 5: We appreciated this recommendation, we added two sentences at the end of section 3 (see below).
Five parallel measurements were conducted and averaged for each final result. The standard deviation was indicated by error bars.
